# *BRCA1* Intragenic Duplication Combined with a Likely Pathogenic *TP53* Variant in a Patient with Triple-Negative Breast Cancer: Clinical Risk and Management

**DOI:** 10.3390/ijms25116274

**Published:** 2024-06-06

**Authors:** Vuthy Ea, Claudine Berthozat, Hélène Dreyfus, Clémentine Legrand, Estelle Rousselet, Magalie Peysselon, Laura Baudet, Guillaume Martinez, Charles Coutton, Marie Bidart

**Affiliations:** 1UM Génétique Moléculaire: Maladies Héréditaires et Oncologie, University Hospital Grenoble Alpes, 38000 Grenoble, France; eavuthy@gmail.com; 2INSERM U1209, CNRS UMR 5309, Institute for Advanced Biosciences, Grenoble Alpes University, 38000 Grenoble, France; gmartinez@chu-grenoble.fr (G.M.); ccoutton@chu-grenoble.fr (C.C.); 3Department of Medical Oncology, Cancer and Blood Diseases, Grenoble Alpes University Hospital, 38000 Grenoble, France; cberthozat@chu-grenoble.fr; 4Genetic Service, Department of Genetics and Procreation, University Hospital Grenoble Alpes, 38000 Grenoble, France; hdreyfus@chu-grenoble.fr (H.D.); clegrand2@chu-grenoble.fr (C.L.); erousselet@chu-grenoble.fr (E.R.); mpeysselon@chu-grenoble.fr (M.P.); lbaudet@chu-grenoble.fr (L.B.); 5UM de Génétique Chromosomique, University Hospital Grenoble Alpes, 38000 Grenoble, France

**Keywords:** hereditary breast and ovarian cancer syndrome (HBOC), multi-gene panel testing, double heterozygotes, genetic counseling, exon duplication

## Abstract

For patients with hereditary breast and ovarian cancer, the probability of carrying two pathogenic variants (PVs) in dominant cancer-predisposing genes is rare. Using targeted next-generation sequencing (NGS), we investigated a 49-year-old Caucasian woman who developed a highly aggressive breast tumor. Our analyses identified an intragenic germline heterozygous duplication in *BRCA1* with an additional likely PV in the *TP53* gene. The *BRCA1* variant was confirmed by multiplex ligation probe amplification (MLPA), and genomic breakpoints were characterized at the nucleotide level (c.135-2578_442-1104dup). mRNA extracted from lymphocytes was amplified by RT-PCR and then Sanger sequenced, revealing a tandem duplication r.135_441dup; p.(Gln148Ilefs*20). This duplication results in the synthesis of a truncated and, most likely, nonfunctional protein. Following functional studies, the *TP53* exon 5 c.472C > T; p.(Arg158Cys) missense variant was classified as likely pathogenic by the Li-Fraumeni Syndrome (LFS) working group. This type of unexpected association will be increasingly identified in the future, with the switch from targeted BRCA sequencing to hereditary breast and ovarian cancer (HBOC) panel sequencing, raising the question of how these patients should be managed. It is therefore important to record and investigate these rare double-heterozygous genotypes.

## 1. Introduction

Breast cancer (BC) is the most common cancer in the world; an estimated 2.3 million new cases were diagnosed in 2022 [1]. Around 10% of cases arise in patients with an autosomal dominant predisposition, termed hereditary breast and ovarian cancer syndrome (HBOC). This syndrome is mainly caused by germline pathogenic variants (PVs) affecting *BRCA1* (OMIM #604370) or *BRCA2* (OMIM #612555) genes. *BRCA1* is the most frequently altered gene in HBOC, with carriers of PV having a cumulative risk for BC up to age 80 years estimated at 65–79% [2]. This tumor-suppressor gene, located at chromosome 17q21.31, has 23 exons and encodes a protein mostly known for its role in the repair of DNA double-strand breaks via homologous recombination. Thousands of *BRCA1* PVs have been reported, most of which create premature stop codons leading to the production of truncated proteins and reduced expression through nonsense-mediated mRNA decay [3]. Notably, extensive rearrangements account for more than 10% of *BRCA1* alterations [4].

Other genes such as *CDH1, PALB2, PTEN, RAD51C, RAD51D, MLH1, MSH2, MSH6, PMS2, EPCAM* and *TP53* also have a high or moderate penetrance in BC. As a result, broad genetic screening including these 11 genes has progressively replaced targeted *BRCA* testing in hereditary breast and ovarian cancer (HBOC) [5]. Germline PVs in *TP53* have been associated with a syndrome—Li-Fraumeni syndrome (LFS, OMIM #151623). This cancer predisposition syndrome is associated with various childhood- or adult-onset malignancies, with early-onset BC matching the 2015 Chompret criteria for LFS when diagnosed before age 31 [6]. However, following the expansion of NGS using the full HBOC panel, *TP53* variants are now detected in a much wider variety of phenotypes. Overall, this shift has implications for both genetic counseling and theranostic decisions. Indeed, germline *BRCA* carriers can now benefit from treatment with poly (ADP-ribose) polymerase inhibitors (PARPi) for ovarian, breast, pancreatic and prostate cancer [7,8,9,10]. Therefore, it is crucial to accurately assess variant pathogenicity. This assessment generally follows the guidelines from the American College of Medical Genetics and Genomics and the Association for Molecular Pathology (ACMG-AMP) [11], which sometimes require additional follow-up [12].

Although unusual, co-occurrence of *BRCA* alterations with rare variants disrupting other DNA-repair genes has been described elsewhere [13,14], but its association with an earlier onset of BC remains a subject of debate. Variant co-occurrence raises questions on the clinical management of the proband and first-degree relatives, mainly because they face a potentially higher risk than if they had only a single PV.

Here, we report the characterization of a *BRCA1* duplication of exons 4 to 6 at the DNA and RNA levels, associated with a likely PV in the *TP53* gene. These mutations were discovered after NGS in a patient with highly aggressive triple-negative BC (TNBC).

## 2. Results

### 2.1. Patient

The proband was diagnosed at 49 years of age with TNBC on the left breast and axillary lymphadenopathy. She received neoadjuvant chemotherapy (dose-dense doxorubicin–cyclophosphamide (AC) X4_ 12 paclitaxel) from May 2021 to November 2021, followed by total mastectomy and axillary lymph node dissection in December 2021. Histological analyses revealed an aggressive tumor classified as ypT3 ypN3a (capsular rupture was noted in 26 of 31 lymph nodes) and, thus, RCB III. Adjuvant radiotherapy was administered during February and March 2022. A subsequent assessment of extension identified no metastatic lesions. Given her young age, the patient was referred to the clinical genetics department of Grenoble University Hospital. Her maternal family history comprised an aunt diagnosed with hormone-receptor-positive (HR+) BC at age 65 and a grandfather who died aged 79 from prostate cancer. On the paternal branch, her father developed gastric cancer after age 70, an uncle underwent treatment for colorectal cancer at age 69, and her grandmother died at age 38, possibly from hepatic neoplasia. An HBOC genetic analysis was requested because of the patient’s own diagnosis of TNBC before age 50 (Figure 1). This analysis identified a duplication of *BRCA1* exons 4 to 6, as well as a heterozygous missense variant in *TP53* exon 5 (c.472C > T; p.(Arg158Cys)). In line with current recommendations on the interpretation of intragenic duplications, we further characterized the *BRCA1* duplication. While awaiting its definitive characterization, it has been classified likely PV (ACMG classification [11]). As a result, the patient was able to benefit from treatment with a PARP inhibitor from April 2022 to January 2023. Concomitantly, preventive mastectomy of the right breast revealed a TNBC (2 mm) without vascular emboli or perineural sheathing, classified pT1a Nx. Due to the *TP53* mutation, in January 2023, a full-body magnetic resonance imaging (full-body MRI) was performed. Spinal bone lesions were discovered and biopsies confirmed metastatic relapse, despite the PARP inhibitor treatment. During March and May 2023, the patient received a first-line therapy for metastatic BC. Her general condition deteriorated suddenly and she died in June 2023, i.e., 4 months after the diagnosis of metastatic relapse. Chemo-resistance, metastatic relapse just 1 year after complete surgical management and survival of 4 months in the metastatic phase confirm that this patient had an extremely aggressive tumor.

### 2.2. MLPA and Breakpoints

Using MLPA on genomic DNA from the proband, the *BRCA1* duplication was confirmed to be in a heterozygous state (Figure 2A). To characterize the breakpoints, we performed a long-range PCR with forward and reverse primers located in *BRCA1* exon 6 and exon 4, respectively, applying a primer walking strategy to narrow down the breakpoint for the recombination (Figure 2B). Sanger sequencing identified a 25 bp portion overlapping introns 6 and 3, which led to a duplication spanning 8128 bp. In line with The Human Genome Variation Society (HGVS) nomenclature, this duplication was described as g.43101009_43109136dup at the DNA level (hg38) and c.135-2578_442-1104dup at the cDNA level (Figure 2C). RepeatMasker indicated that this 25 bp region was shared by two *Alu* elements oriented in the same direction and flanking the duplication breakpoint in introns 6 and 3: *AluSx* and *AluJr*, respectively, with a partial overlap spanning 66 bp (Figure 2D).

### 2.3. RNA Analysis

Finally, we investigated the impact of duplication of *BRCA1* exons 4 to 6 at the RNA level by performing RT-PCR on the proband’s RNA. We designed primers that would specifically amplify a tandem duplication involving exon 6 (reverse and forward primers both in exon 6 but in opposite directions). A single band of size corresponding to a duplication from exons 4 to 6 (306 bp) was obtained (Appendix A). Sanger sequencing of this RT-PCR product confirmed that the duplication occurred in tandem and in direct orientation. Thus, the consequence of this duplication can be described as r.135_441dup, and it is predicted to produce a truncated protein p.(Gln148Ilefs*20) using Alamut Visual Plus v.1.9 (Interactive Biosoftware)). We also identified less abundant, shorter amplification products, which could correspond to alternative splicing of exon 4 (known as Δ5q) and skipping of the duplicated exon 4 (Appendix A). Ultimately, the characterization of this *BRCA1* alteration allowed us to offer the patient treatment with PARPi.

## 3. Discussion

In this article, we report a germline alteration of *BRCA1*, with duplication of exons 4 to 6, which co-occurred with a likely PV in the *TP53* gene in a patient with aggressive BC. The duplication identified was verified and validated by MLPA and the breakpoints flanking the duplicated sequence were characterized at the nucleotide level. Further characterization of this variant at the RNA level revealed the production of an abnormal transcript leading to the translation of a truncated protein.

The French OncoGenetics database (FrOG_db) reports duplication of *BRCA1* exons 4 to 6 in 10 patients [15] but no information on the genomic orientation of these duplications is available. However, the genomic duplication of exons 4 to 6 has been described three times in the literature: in one patient referred to a French hospital [16], in one patient from Stanford (Chinese ethnicity) [17], and in another patient studied in a Portuguese hospital [18]. The breakpoints were mapped in all three cases and matched to three distinct pairs of Alu elements, which is not surprising, since both introns 3 and 5 are long—intron 3 is the longest *BRCA1* intron (9192 bp)—and contain numerous Alu repeats. Based on this mapping, the duplications arose independently but share the same outcome, i.e., a *BRCA1* loss-of-function linked to HBOC syndrome. In all three cases, the duplications occurred in tandem with a direct orientation; consequently, they are classified as pathogenic. The genomic breakpoints of these copy number variations (CNVs) must be determined to interpret how they affect genes and correlate with phenotypes [19]. Interestingly, the breakpoints we described matched those for the French patient described in the literature [16], suggesting that this duplication could be a recurrent Alu element-mediated CNV. It would be interesting to investigate whether these breakpoints are shared with patients identified in other French centers. The results of our RNA analysis of this *BRCA1* duplication further support its pathogenicity, which is consistent with the diagnosis of TNBC before age 50 [20]. Classification of this variant as PV allowed us to implement appropriate clinical management and genetic counseling for at-risk relatives.

Unlike BRCA-associated BCs, those linked to germline PV in the *TP53* gene tend to be hormone-receptor- and/or HER2-positive [21]. For the patient described here, the contribution of the TP53 p.(Arg158Cys) variant is unclear. We and the French LFS working group classified it as likely pathogenic, according to the ACMG-AMP and ClinGen recommendations based on PM2, PM5 and PP3_moderate criteria [22]. PS3_moderate could arguably be applied because of conflicting evidence, as p.(Arg158Cys) was partially functional in a Kato assay [23] and functionally abnormal in a Rouen assay [24] but remained functional in a Kotler assay [25]. This classification would not be sufficient to upgrade this variant to PV. And, indeed, the proband does not meet the 2015 Chompret criteria [6]. However, partly functional TP53 variants may be linked to later-onset cancers compared to loss-of-function variants [26]. Specifically, this variant has been identified in women who developed BC at age 49 and 53 years [27,28]. It was also described in a 39-year-old patient with adrenocortical cancer and a history of testicular cancer at age 20, who presented a somatic loss of heterozygosity and p53-negative immunohistochemistry staining [29]. Segregation analysis was useful in order to test a de novo or mosaicism hypothesis. Mosaicism occurs in at least 14% of patients with LFS [30].

The proband’s unaffected mother was subsequently found to have the same germline *BRCA* mutation. The proband’s father had a personal history of gastric cancer and tested positive for the familial TP53 mutation. The proband’s unaffected sister tested positive for both the *TP53* and *BRCA1* mutations. For relatives, systematic presymptomatic testing is recommended, with complete annual medical follow-up for carriers. Detection of the germline likely PV in the TP53 gene has theranostic implications. The balance between the risk of recurrence in the absence of chemo- or radiotherapy and the risk of secondary primary tumor induced by the treatment [31] has been discussed by a multi-disciplinary team.

In conclusion, in this patient with TNBC, using both DNA and RNA analyses, we characterized a tandem and direct intragenic duplication in BRCA1, co-occurring with a likely pathogenic *TP53* variant. To the best of our knowledge, this is the first description of a *BRCA1* PV associated with a likely PV in the *TP53* gene. Data on double-heterozygous PV are limited and, as a result, prospective follow-up of these rare patients and their families is not yet well established. The age of onset with very poor prognosis raises the question of how double-heterozygous patients should be managed. It remains important to reference and investigate these rare double-heterozygous genotypes.

## 4. Material and Methods

### 4.1. Subject and DNA Extraction

The patient, 49 years old, was enrolled from the Cancer and Blood Diseases, Department of Medical Oncology of Grenoble Alpes University Hospital, France. She was diagnosed with TNBC and had a positive family history for breast, colon, stomach and hepatic cancers. The patient gave informed consent and was tested following genetic counseling. Total genomic DNA was extracted from a peripheral blood sample in EDTA using the NucleoSpin Blood L Midi kit (Macherey-Nagel, Düren, Germany) by applying an automated procedure implemented on a JANUS^®^ workstation (PerkinElmer, Waltham, MA, USA). DNA was quantified using a nanodrop 2000 (Thermo Fisher Scientific, Invitrogen, Villebon sur Yvette, France).

### 4.2. NGS Analysis and Confirmation

The patient’s DNA was analyzed using the CE-IVD Hereditary Cancer Solution assay (SOPHiA GENETICS, Saint-Sulpice, Switzerland), with parallel sequencing of 27 genes (*ATM*, *APC*, *BARD1*, *BRCA1*, *BRCA2*, *BRIP1*, *CDH1*, *CHEK2*, *EPCAM*, *FAM175A*, *MLH1*, *MRE11A*, *MSH2*, *MSH6*, *MUTYH*, *NBN*, *PALB2*, *PIK3CA*, *PMS2*, *PMS2CL*, *PTEN*, *RAD50*, *RAD51C*, *RAD51D*, *STK11*, *TP53* and *XRCC2*) according to the manufacturer’s recommendations. Briefly, 200 ng of gDNA was enzymatically digested and underwent end repair and A-tailing. Libraries were quantified using a Qubit dsDNA HS Assay Kit on a Qubit 2.0 fluorometer (Thermo Fisher Scientific, Invitrogen, Villebon sur Yvette, France). The library size was verified using capillary electrophoresis (2200 TapeStation, Agilent Technologies, Santa Clara, CA, USA). Sequencing was performed on Illumina NextSeq 500 (Illumina, San Diego, CA, USA). Data were processed on the SOPHiA DDM™ platform (v5.10; SOPHiA GENETICS). Sophia DDM can detect single-nucleotide variants (SNVs), indels, copy number variations (CNVs) and Alu insertions. Pathogenic and likely pathogenic variants were classified with a multifactorial model based on ACMG guidelines [12;22]. Variants were confirmed on a second sample by a second appropriate technique. A missense variant in the TP53 gene was confirmed by Sanger sequencing on a 3500xL Genetic Analyzer (Thermo Fisher Scientific, Waltham, MA, USA). Duplication of *BRCA1* exons 4–6 was confirmed by multiplex ligation-dependent probe amplification (MLPA) using the SALSA MLPA P002 BRCA1 probe mix (MRC Holland, Amsterdam, The Netherlands) according to the manufacturer’s recommendations. Reference sequences used were NM_007294.4 (*BRCA1*) and NM_000546.6 (*TP53*).

### 4.3. PCR and Sanger Sequencing

The *BRCA1* duplication was hypothesized to occur in tandem. To verify this, a long-range PCR (LR-PCR) was performed using primers located in *BRCA1* exon 6 (forward: CAGAGTGAACCCGAAAATCC) and exon 4 (reverse: GGTTGAGAAGTTTCAGCATGC). Since the LR-PCR product could be up to ~13,400 bp, LongAmp^®^ Taq DNA Polymerase (New England Biolabs, Ipswich, MA, USA) was used with a primer walking strategy to narrow down the recombination breakpoint. The duplication junction was amplified with specific primers located in intron 6 (forward: TGGGCTTTTAAATACTCGTTCC) and intron 3 (reverse: GCAATGAGCTGAGAAGAGTGC). PCR products were sequenced using BigDye™ Terminator v3.1. All reagents were used according to the manufacturer’s recommendations.

### 4.4. RNA Analysis

Total RNA was extracted from a fresh blood sample and from an EBV-transformed lymphoblastoid cell line, with and without NMD inhibitor (puromycin treatment) using phenol-chloroform protocol. RNA was reverse transcribed into cDNA using Transcriptor Reverse Transcriptase kit (Roche Diagnostics, Indianapolis, IN, USA). Touchdown PCR was performed using primers located in BRCA1 exon 6 (forward: ACAGAGTGAACCCGAAAATC and reverse: GAAGTCTTTTGGCACGGTTT). PCR products were sequenced using BigDye™ Terminator v3.1 on a 3500xL Genetic Analyzer.

## Figures and Tables

**Figure 1 ijms-25-06274-f001:**
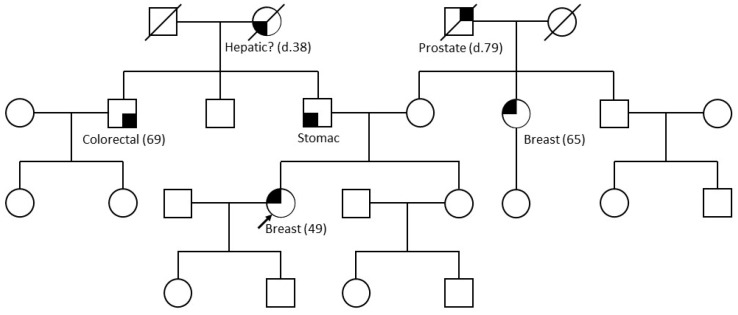
Pedigree of the patient referred for HBOC gene sequencing, indicating known cancer events in her family.

**Figure 2 ijms-25-06274-f002:**
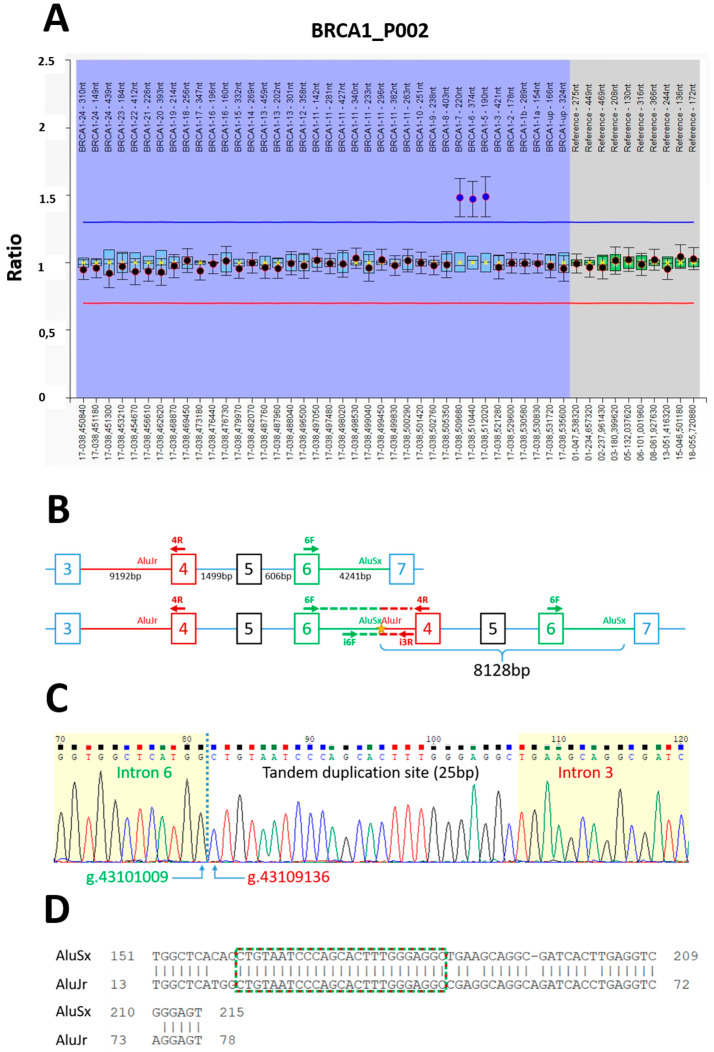
*BRCA1* exons 4–6 duplication and breakpoint identification. (**A**) Duplication confirmation by MLPA analysis. Exons 4–6 are referred to as 5–7 in an ancient nomenclature. (**B**) Schematic representation of the *BRCA1* normal and duplicated alleles with *Alu* regions involved in this duplication. Primers used for the long-range PCR and the duplication specific PCR are indicated by horizontal arrows, with their expected PCR products in dashed lines. For the sake of simplicity, elements linked to intron 3/exon 4 are colored in red, while those linked to exon/intron 6 are in green. (**C**) Electropherogram of the breakpoint sequence (forward from i6F) and positions (hg38), showing the tandem duplication site common to introns 6 and 3 (white background). (**D**) Partial alignment of the two *Alu* elements around the breakpoint region, illustrating their partial homology. Tandem duplication box is framed in green and red.

## Data Availability

Dataset available on request from the authors.

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
