# Peer review of "BRCA1 Intragenic Duplication Combined with a Likely Pathogenic TP53 Variant in a Patient with Triple-Negative Breast Cancer: Clinical Risk and Management"

_ijms, 2024, doi:10.3390/ijms25116274_

Round 1

Reviewer 1 Report

Comments and Suggestions for Authors

In this study, authors reported BRCA1 duplication in a patient with Triple-Negative Breast Cancer. They used NGS to detect variations at DNA and RNA levels. Below are a number of issues that the authors shall address or revise:

1. Authors should check figures and figure legends. Figure 1 was missing in the manuscript, and Figure 2 should be Figure 1. Two figures appeared in the Figure S1.

2. Only one case was reported with the BRCA1 duplication and TP53 variant in this report. I wonder whether this kind of variation can be found in other cases that also develop aggressive cancer. Authors can use some public databases or FFPE samples in the hospital. More than one case should be used to support the prediction.

3. I wonder whether authors can get more genetic information of other patients in this patient’s family and whether her kids or siblings can benefit from this report.

Author Response

1. Authors should check figures and figure legends. Figure 1 was missing in the manuscript, and Figure 2 should be Figure 1. Two figures appeared in the Figure S1. Corrected 

2. Only one case was reported with the BRCA1 duplication and TP53 variant in this report. I wonder whether this kind of variation can be found in other cases that also develop aggressive cancer. Authors can use some public databases or FFPE samples in the hospital. More than one case should be used to support the prediction. Hereditary breast cancer (BC) corresponds to 5% of all BC. If co-occurence of the BRCA and TP53 pathogenic variant  is frequently found on tumour analysis, there are few reports in the literature on the co-occurrence and the cancer phenotypes associated with double heterozygous pathogenetic variants in cancer predisposing genes. In our single institution, 1500 women with early onset BC and/or a family history of cancer were tested by a germline multi-gene hereditary cancer panel : (HBOC). We have identified a single DH case (TP53/BRCA1). This is a rare event, and there are few reports on its impact on cancer risk. 

3. I wonder whether authors can get more genetic information of other patients in this patient’s family and whether her kids or siblings can benefit from this report. The known results of segregation have been mentioned in the discussion. We have no other results. For example, while we have recommended a test for her kids (13 and 16 years old), this has not yet been carried out by the family. Depending on the genetic results, a Li-Fraumeni follow-up may be set up for related individuals.

Reviewer 2 Report

Comments and Suggestions for Authors

This is an interesting and well-written manuscript that reports an association between BRCA1 duplication of exons 4 to 6 with likely PV in the TP53 gene in a patient with highly aggressive breast cancer. The Introduction describes the study in the broad context highlighting the importance of the study. However, I suggest including more information in Materials and Methods to make clearer the procedures for DNA and RNA extraction. Some results are incomplete. Discussion is clear and includes interesting supporting references. Finally, references should be corrected. In addition, I suggest to consider next few minor comments:.

Materials and Methods:

-       Section 4.1. I suggest including more descriptive information about the patient. Also, I suggest describing briefly the technique or procedures used for DNA extraction and quantification.

-       Section 4.4. I suggest describing briefly the technique or procedures used for RNA extraction.

-       Line 210: What was(were) the procedure(s) used to identify the missense variant in the TP53 gene?

-       Note: All bioinformatics resources reported in the study should include the website address and the date of accession.

Results:

-       Line 85: Does the HBOC genetic analysis performed in this study?. If so, please describe it in Materials and Methods section.

-       Line 87: How was the missense variant in TP53 detected?. Please explain briefly.

-       Line 90: Please include the reference for the “ACMG classification”.

-       Line 94: Does the full-body MRI analysis performed in this study?. If so, please describe it in Materials and Methods section.

-       Line 101: Figure 1 is missing.

-       Line 116: Please clarify in the figure what correspond to A, B, C and D.

-       Line 131: Figure S1 A is missing.

-       Line 136: Figure S1 B is missing. 

References:

-       Please correct references grammar by following the guidelines described in the “Instructions for Authors”.

-       References should be cited in the manuscript as progressive numbers within a square bracket instead of a round bracket.

Author Response

To consider next few minor comments:.

Materials and Methods:

-       Section 4.1. I suggest including more descriptive information about the patient. Also, I suggest describing briefly the technique or procedures used for DNA extraction and quantification. Done : line 207 to 209 and 214-215

-       Section 4.4. I suggest describing briefly the technique or procedures used for RNA extraction. Done : line 251 to 253

-       Line 210: What was(were) the procedure(s) used to identify the missense variant in the TP53 gene? Done line 221 to 233

-       Note: All bioinformatics resources reported in the study should include the website address and the date of accession. Added

Results:

-       Line 85: Does the HBOC genetic analysis performed in this study?. If so, please describe it in Materials and Methods section. Done line 221 to 233

-       Line 87: How was the missense variant in TP53 detected?. Please explain briefly. Described in conclusion line 172_179

-       Line 90: Please include the reference for the “ACMG classification”. Done [11]

-       Line 94: Does the full-body MRI analysis performed in this study?. If so, please describe it in Materials and Methods section: Full-body MRI has been carried out in our hospital, so we do not have this information at our disposal.

-       Line 101: Figure 1 is missing. Added

-       Line 116: Please clarify in the figure what correspond to A, B, C and D. Done

-       Line 131: Figure S1 A is missing. Added

-       Line 136: Figure S1 B is missing. Added

References:

-       Please correct references grammar by following the guidelines described in the “Instructions for Authors”. Corrected

-       References should be cited in the manuscript as progressive numbers within a square bracket instead of a round bracket. Corrected

Reviewer 3 Report

Comments and Suggestions for Authors

The authors present the article entitled BRCA1 Intragenic Duplication Combined with a Likely Pathogenic TP53 Variant in a Patient with Triple-Negative Breast Cancer: Clinical Risk and Management. The paper reports the finding of two genetic variants, in the BRCA1 and TP53 genes, present in a patient with hereditary breast and ovarian cancer syndrome.
The text highlights how rare are reports where two variants are detected in high-penetrance genes associated with breast cancer, contrasting this with the finding of their patient.
The article is interesting and its methodological strategy well designed, however, I have some observations.

1. I consider that the introduction should cover a little more about the findings of multiple genetic variants associated with breast cancer or different types of hereditary cancer syndromes, mainly with genes of high or moderate penetrance.

2. In the results section, the authors write "This analysis identified a duplication of BRCA1 exons 4 to 6, as well as a missense variant in TP53 exon 5 (c.472C>T; p.(Arg158Cys))... ", I consider that it should be pointed out that it was through NGS analysis and mention if they are heterozygous, since it is not clear.

3. The authors do not mention in the results whether a family segregation study was carried out, which in this case is of utmost importance to detect possible carriers of the variants. In this sense, the authors mention a little about this in the discussion section.

4. In the results section, the pedigree figure and its legend are not in the same position. You can also only read the legend of Figure 2, but the figure is not there.

5. In the results section, RNA analysis, the authors mention that the production of a truncated protein is predicted. I consider that it should be mentioned how they determined it or what type of tools they used to make this statement.

6. I believe that in the discussion the authors should argue a little more about the most probable diagnosis of the patient, that is, HBOC or Li-Fraumeni syndrome, especially due to the type of cancer presented by the family members.

7. In the methodology, I consider that the description of MLPA and RNA analysis should be expanded.

8. Improve the quality of the images in your figures because some look poorly defined.

8. The figure of the agarose gel together with that of the duplicated regions should be improved to know well what each lane called "patient" refers to, to know which exon scheme it is associated with.

9. Arrange the figures and their legends in the corresponding place.

Comments on the Quality of English Language

Moderate editing of English language required. It is recommended that the writing be reviewed and corrected

Author Response

  1. I consider that the introduction should cover a little more about the findings of multiple genetic variants associated with breast cancer or different types of hereditary cancer syndromes, mainly with genes of high or moderate penetrance. We have limited our introduction to the BRCA and TP53 genes of interest for our case report. In fact, there are other genes, cited in the article, associated or not with other syndomes (Lynch, Cowden, etc) but we are limited by the number of characters.

    2. In the results section, the authors write "This analysis identified a duplication of BRCA1 exons 4 to 6, as well as a missense variant in TP53 exon 5 (c.472C>T; p.(Arg158Cys))... ", I consider that it should be pointed out that it was through NGS analysis and mention if they are heterozygous, since it is not clear. Added in results and Mat Met

    3. The authors do not mention in the results whether a family segregation study was carried out, which in this case is of utmost importance to detect possible carriers of the variants. In this sense, the authors mention a little about this in the discussion section. The known results of segregation have been mentioned in the discussion. We have no other results. For example, while we have recommended a test for children (13 and 16 years old), this has not yet been carried out by the family

    4. In the results section, the pedigree figure and its legend are not in the same position. You can also only read the legend of Figure 2, but the figure is not there. Corrected

    5. In the results section, RNA analysis, the authors mention that the production of a truncated protein is predicted. I consider that it should be mentioned how they determined it or what type of tools they used to make this statement. Added line 140 

    6. I believe that in the discussion the authors should argue a little more about the most probable diagnosis of the patient, that is, HBOC or Li-Fraumeni syndrome, especially due to the type of cancer presented by the family members. The proband does not meet the 2015 Chompret criteria 

    7. In the methodology, I consider that the description of MLPA and RNA analysis should be expanded. Added line 234-237 and 251_254

    8. Improve the quality of the images in your figures because some look poorly defined. Corrected

    8. The figure of the agarose gel together with that of the duplicated regions should be improved to know well what each lane called "patient" refers to, to know which exon scheme it is associated with. Done (see Figue 2)

  2. 9. Arrange the figures and their legends in the corresponding place. Corrected

Reviewer 4 Report

Comments and Suggestions for Authors

Review of (ijms-2993500)

BRCA1 Intragenic Duplication Combined with a Likely 2 Pathogenic TP53 Variant in a Patient with Triple-Negative 3 Breast Cancer: Clinical Risk and Management

In this manuscript, the authors presented a case of very aggressive triple negative breast cancer with an intragenic germline heterozygous duplication in BRCA1 with an additional likely pathogenic variance in the TP53 gene. They used in their genetic analysis various techniques including Next generation sequencing, MLPA, RT-PCR and sanger sequencing to fully describe the detected mutations.

Although the idea of studying BRCA1 and TP53 in breast cancer is not new. This manuscript described the synergistic effect of both gene is a novel idea which should help in better understanding in TNBC biology. 

General comment about the writing: The paper is well written and very easy to follow.

·      Please review all the manuscript to make sure that you spelled out all the abbreviations the first time they are mentioned in the body of the manuscript. 

Introduction: well written. However, for the statistics of breast cancer mentioned in (Line 35, 36) need to be updated as there is a more recent global cancer statistics (2022) in which lung cancer and the breast is the second common worldwide.

Results: 

The clinical history and the description of the case are comprehensive.

Adding the strong family history of various types of cancer was important addition in the case description. 

The authors provided comprehensive molecular characterization of the BRCA1 duplication.

Minor comments:

·      Please refer to figure 1 in the text.

·      Review the place of the legend in the paper as Figure 2 legend in placed under figure 1.

·      Figure 2 legend (line 125), you mention red box, but no red boxes found in the figure.

·      Figure 2A, please increase the resolution of the picture to make the writing readable.

Author Response

Statistics of breast cancer mentioned in (Line 35, 36) need to be updated as there is a more recent global cancer statistics (2022) in which lung cancer and the breast is the second common worldwide : OK Updated

Minor comments:

  • Please refer to figure 1 in the text  : Done line 86
  • Review the place of the legend in the paper as Figure 2 legend in placed under figure 1 : Done
  • Figure 2 legend (line 125), you mention red box, but no red boxes found in the figure. It's a box that's both green and red, I've improved the resolution so we should be able to see it better.
  • Figure 2A, please increase the resolution of the picture to make the writing readable : Done
  •     Please review all the manuscript to make sure that you spelled out all the abbreviations the first time they are mentioned in the body of the manuscript : Done